# Effect of Rot-, Fire-, and Water-Retardant Treatments on Jute Fiber and Their Associated Thermoplastic Composites: A Study by FTIR

**DOI:** 10.3390/polym13152571

**Published:** 2021-08-01

**Authors:** Sweety Shahinur, Mahbub Hasan, Qumrul Ahsan, Nayer Sultana, Zakaria Ahmed, Julfikar Haider

**Affiliations:** 1Department of Testing and Standardization, Bangladesh Jute Research Institute, Manik Mia Avenue, Dhaka 1207, Bangladesh; 170178sweety@gmail.com; 2Department of Materials and Metallurgical Engineering, Bangladesh University of Engineering and Technology, Dhaka 1000, Bangladesh; mahbubh@mme.buet.ac.bd; 3Department of Mechanical and Production Engineering, Ahsanullah University of Science and Technology, Dhaka 1208, Bangladesh; qumrul.mpe@aust.edu; 4Pilot Plant and Processing Division, Bangladesh Jute Research Institute, Manik Mia Avenue, Dhaka 1207, Bangladesh; Nayersultana99@gmail.com; 5Department of Microbiology, Bangladesh Jute Research Institute, Manik Mia Avenue, Dhaka 1207, Bangladesh; zakariaahmed70@gmail.com; 6Advanced Materials and Surface Engineering (AMSE) Research Centre, Manchester Metropolitan University, Manchester M1 5GD, UK

**Keywords:** rot-retardant, flame-retardant, water-retardant, natural fiber, jute, thermoplastic composite, FTIR

## Abstract

Natural renewable materials can play a big role in reducing the consumption of synthetic materials for environmental sustainability. Natural fiber-reinforced composites have attracted significant research and commercial importance due to their versatile characteristics and multi-dimensional applications. As the natural materials are easily rotten, flammable, and moisture absorbent, they require additional chemical modification for use in sustainable product development. In the present research, jute fibers were treated with rot-, fire-, and water-retardant chemicals and their corresponding polymer composites were fabricated using a compression molding technique. To identify the effects of the chemical treatments on the jute fiber and their polymeric composites, a Fourier transformed infrared radiation (FTIR) study was conducted and the results were analyzed. The presence of various chemicals in the post-treated fibers and the associated composites were identified through the FTIR analysis. The varying weight percentage of the chemicals used for treating the fibers affected the physio-mechanical properties of the fiber as well as their composites. From the FTIR analysis, it was concluded that crystallinity increased with the chemical concentration of the treatment which could be contributed to the improvement in their mechanical performance. This study provides valuable information for both academia and industry on the effect of various chemical treatments of the jute fiber for improved product development.

## 1. Introduction

Jute is a natural fiber, an environmentally sustainable material, that absorbs CO_2_ during the primary material production as it is collected from plants, and is 100% biodegradable. Furthermore, it is an annual crop with a cultivation time ranging between 110 and 120 days; it contains 80–75% holo-cellulose and approximately 11–15% lignin. Significant research efforts over the past few decades have made considerable progress in jute fiber-reinforced polymer composites [1,2,3,4,5,6]. Particular attention has been devoted to enhancing the interfacial strength of jute fiber with the matrix materials by treating the fiber with various chemicals [7,8] to achieve physical and chemical modifications with or without cross-linking [9,10]. Different types of treatments on the jute fiber and the corresponding impacts in the resulting composite are summarized in Figure 1 [11,12,13].

Meanwhile, one of the common uses of jute fiber is in fiber-reinforced composites. For the composite material, jute in different forms (long, short, particle, or woven) was incorporated in matrix materials with different forms (liquid, granule, sheet, or plate) as schematically presented in Figure 2. Bonding/interfacial strength has been improved through different chemical treatments such as NaOH [14], CH_3_COOH [8], H_2_O_2_, HCl, graphene [15], citric acid [8], NaCl, HCOOH, NaHCO_3_, and distilled water [16]. Some studies demonstrated that the treated jute fiber showed better performance in specific fields (e.g., household accessories, footwear additive, car parts, roof tiles, and tank) [17]. Similarly, other studies showed that the treated fiber with radioactivation [9,10] also improved performance in various fields like electrical wire, roofing material like celling, and structural materials like beams and panels.

Furthermore, other types of treatments (bleaching, scouring, mercerization) [18] have also contributed to better bonding between the jute fiber and polymeric matrix material. Good bonding makes a good composite product. However, another major challenge of the jute-based composite/product in applications is that after encountering soil, fire, and water, they easily rot, catch fire, and absorb moisture. Thus, they should be bacteria-retardant, flame-retardant, and moisture-retardant to enhance their life span. It was reported that for anti-bacterial treatment, Cu salt [4], Cl bleaching, and amino-aldehyde resins were used whereas for anti-flaming treatment, NH_4_ salt [19], benzoxazines [20], halogen [21], and fluoro resin powder [22] were used. For water retardant treatment, PVC was used as it had no hydrogen bond [23]. Considerable attention was given by the researchers to characterizing the treated fibers before incorporating them into the matrix material to fabricate the composites [5]. To support their findings, fibers were characterized by X-ray diffraction (XRD), thermogravimetric analysis (TGA), differential scanning calorimetry (DSC), dynamic mechanical analysis (DMA), atomic force microscope (AFM), infrared spectroscopy (IR), and Fourier transformed infrared radiation (FTIR) to identify changes in the percentage of cellulose, presence of functional group, and esterification [24,25,26]. The identification of the foreign particles on the treated fiber may not only give information to the affinity of the jute fiber with the chemicals but also help to predict its effects on the fiber characteristics. This is essential in making an informed decision for jute-based product development. Furthermore, during chemical treatment, the mechanism of chemical bond creation, the existence of single, double, and triple bonds in the fiber, the change in the fiber’s structure, and the relationship between the chemical concentration and physio-chemical properties of the fiber [4] are not fully understood. Furthermore, how the change in the fiber structure affects its strength or the resulted composite needs to be explored. FTIR study could help reveal the hidden information. FTIR data show that a peak shifting left or right indicates an increase in density of the material, which is related to the crystallinity. The appearance of an extra peak means extra material is present in the fiber [27]. This type of information will help to identify the presence of foreign particles in the fiber. Previous researchers studied chemically treated wood, sisal, kenaf, and hemp fibers using FTIR [28,29]. However, the information of cellulose (particularly holo-cellulose) and other content (cellulose, lignin) [28] distribution, and the influence of chemical treatment on crystallinity [30,31] has not yet been explained for the untreated and treated jute fibers. Therefore, FTIR analysis of the treated jute fiber can provide valuable information for the researchers or process engineers to develop jute-based composite parts or products. Based on the above contemplation, jute fibers treated with three different types of chemicals and their corresponding composites were studied by FTIR. The originality of this research work lies in gaining an understanding of the effects of chemical treatment on the jute fiber characteristics and predicting its influence on the resulting composites.

In order to provide a clear idea about the study, the remainder of this article is organized as follows. The experimental procedure section highlights the fiber collection, fiber treatment, and composite fabrication methods. The results and discussions section presents and discusses the results on the chemical treatments of the jute fiber as well as the composites through FTIR analysis and future recommendations are provided. Finally, the important conclusions are drawn in the conclusion section.

## 2. Materials and Methods

In this study, the raw and treated jute fibers as well as their polymeric composites were characterized through Fourier transform infrared spectrophotometer (FTIR). Step-by-step fiber processing, the fabrication of the composites, and FTIR characterization procedure are described in the next sections.

### 2.1. Fiber Collection and Treatment

Jute fiber named CVL-1 (*Corchorus capsularis* L.) was collected from the Faridpur Regional Station of Bangladesh Jute Research Institute (BJRI), Bangladesh. A schematic diagram of the experimental methodology to investigate the FTIR characteristics of the treated jute fibers and their composites is presented in Figure 3.

The middle portions of the whole jute fibers were cut and considered for three different chemical (T1, T2, and T3) treatments. After treatment, the fiber samples were chopped into a size of approximately 3 mm and incorporated into the granule matrix material maleic anhydride-grafted polypropylene (MAgPP) to fabricate the composite samples with 25% (weight percentage). MA content was optimized to 9 wt.% as recommended by the manufacturer (Merck, Germany). The notation of raw and treated jute fibers with their chemical concentrations (wt.%) in the reinforced polymeric composites are listed in Table 1. Finally, FTIR analysis was performed to characterize the fibers and their reinforced composites.

In Table 1, T1, T2, and T3 stand for rot-, fire-, and water-retardant treatments, respectively, while R, F, and W stand for rot-, fire-, and water-retardant chemical concentrations. Furthermore, JT1-PP, JT2-PP, and JT3-PP stand for rot-, fire-, and water-retardant-treated jute fiber-reinforced composites, respectively. The concentrations of the different chemicals used in this study were selected based on the authors’ pilot study and previously published work. For example, Jafrin et al. found that 10% rot-retardant treatment with CuSO_4_ solution performed better in increasing the longevity of jute-based nursery pots [32]. Furthermore, Khatton et al. [33] demonstrated that 30% fire retardant salt (NH4)_2_·HPO4) produced the best fire-retardant characteristics in jute without significantly compromising the mechanical properties. The literature also suggested that 16% PVC solution showed better response as a water-resistant jute product [34] and that is why 10%, 15%, and 20% chemical concentrations were used in this investigation.

### 2.2. Evaluation of Chemical Treatments

To assess the effect of different chemical treatments on the jute fiber and their polymeric composites, anti-bacterial, flammability, and contact angle tests were performed according to the corresponding ASTM standards such as M07-A8 for anti-bacterial, D-1230-17 for flammability, and ASTM D5946 for contact angle.

#### 2.2.1. Anti-Bacterial Test

Reagent grade nutrient agar (NA) media were used throughout the study to test the antimicrobial properties of the treated samples using the Kirby–Bauer disk diffusion susceptibility test protocol [35,36]. The jute fibers treated with three different concentrations of rot retardant chemical were tested against five different bacterial isolates (*Acinetobacter* sp., *Pseudomonas* sp., *Bacillus cereus, Salmonella* sp., and *E. coli*) where the optical density (OD) of all the isolates were standardized at 600 nm (OD = 1.5 × 10^6^). The bacterial activity was determined after 24 h of incubation at 37 °C temperature. The zones of inhibition were determined by the National Committee for Clinical Laboratories Standard rules [37]. The inhibition zone is defined as the clear zone created around the wells by the antibacterial action. Negative controls were set using sterile water. The zone of inhibition is roughly a circular area around the spot of the antimicrobial agent in which the bacteria colonies do not grow, as shown in Figure 4. The calculated inhibition zone diameter determined if a particular bacterium was susceptible or resistant to the applied antimicrobial agent. The larger diameter indicated a higher resistance against a bacterium and vice versa. To measure the zone of inhibition, a ruler was placed directly across the zone of inhibition through the center. All measurements were executed three times to obtain an average result.

#### 2.2.2. Flammability Test

First, a preliminary test was conducted to determine the fastest burning direction of the fibers. Samples were pre-heated in an oven at 105 ± 3 °C for 30 ± 2 min and placed in a desiccator with anhydrous silica gel to cool for at least 15 min. Five specimens were prepared with a dimension of 50 mm by 150 mm to conduct the flammability test by a flammability tester. The test procedure required that a 16 mm flame was impinged on a specimen mounted at a 45° angle for 1 s. The specimen was allowed to burn to its full length or until the stop thread was broken at a distance of 127 mm. The burned areas of several specimens were averaged, and a class designation was made based on the flammability performance [38] as shown in Figure 5.

#### 2.2.3. Contact Angle Test

To realize the water transmittance on the water-retardant treated fiber surface, droplet shape and size were measured. This was done by photographing droplets of liquid on the treated and raw jute fibers using a SONY CORP, digital still camera (12.1 megapixels, 4× optical zoom, 28 mm wide-angle lens, model no. DSC-W310). One drop of liquid was carefully applied to the substrate using a syringe. Distilled water was used to characterize the fiber based on the contact angle of liquid droplets [39]. The contact angle was measured three times and average values were reported (Figure 6).

### 2.3. FTIR Characterization of Fiber

A digital spectrophotometer (Model Nicolet-380, Wisconsin, USA) was employed to conduct FTIR spectroscopy of the raw and treated jute fibers by following the attenuated total reflectance (ATR) technique. The analyses were run using the KBr pellet technique. The samples were scanned with a transmittance range of 370 to 4000 cm^−1^. Transmittance bands of jute fiber were recognized using the Spectra Base™ databases, an accessible online spectral source from John Wiley & Sons, Inc. In order to further ensure the accuracy of identification, transmittance bands of the jute fiber were also compared with published values in the literature [40].

### 2.4. Composite Fabrication and Characterization

At first, 5 mm chopped jute fiber and MAgPP were weighted and oven-dried at 105 ± 3 °C for 6 h to remove the moisture from the raw materials. Thereafter, granules of PP were heated with fiber (25 wt.%) at 120 °C until they started melting [19]. A stirrer was used to mix the fiber and matrix material to ensure the homogeneous distribution of the fiber as much as possible. When the fiber was stuck with the matrix material, they were poured into the female dice and placed in the hot press machine. At 185 °C, 30 kN pressure was applied on the dice for 10 min. Then, the system was cooled slowly using a water-cooling system. The composite colors with the treated fibers seemed slightly darker compared to the composite with the non-treated jute. This could be related to the change in color in the chemical-treated jute fibers. Furthermore, the processing temperature was 185 °C, which was much lower than the jute degradation temperature as demonstrated in a previous study [19]. Finally, the specimen was carefully removed from the dice and prepared for FTIR analysis by following the method as discussed in Section 2.3.

## 3. Results and Discussions

Important results related to the chemical modifications of the jute fibers and their composites are presented divided into three subsections. The first subsection discusses the verification of the chemical treatment and the second subsection contains the FTIR data analysis of the raw and treated jute fibers. Finally, the third subsection highlights FTIR data analysis of the composites reinforced with untreated and treated jute fibers.

### 3.1. Rot-, Fire-, and Water-Retardant Fiber Characteristics

In all T1 treatments, most of the zones of inhibition developed against the microbes used showed higher diameters than their corresponding controls, thus confirming having antimicrobial properties. It was observed that T1R3 showed higher activity in inhibiting bacterial growth compared to T1R2 and T1R1 against *Acinetobacter* sp., *Bacillus cereus*, and *Pseudomonas* sp., whereas the T1 treated jute fiber showed insignificant antimicrobial activity against *Salmonella* sp. and *E. coli* (Table 2). However, at the highest concentration, an inhibition zone was developed against *E. coli*. Therefore, it was clear that the rot retardant treatment was not effective against all types of bacteria.

Jafrin et al. treated jute fabrics with rot retardant using different concentrations of CuSO_4_ solution for nursery plant pot applications and they showed better performance in soil contact conditions [32]. They found that 10% of CuSO_4_ treatment showed high durability in the soil meaning more anti-bacterial activity. In this study, it was also found that the antibacterial activity increased with the chemical concentration and therefore CuSO_4_ is a good candidate as an antibacterial material for jute products like geo-textiles. Other treatments on jute with natural henna and biopolymer chitosan were also studied by Bhuiyan et al. in order to incorporate the rot-retardant characteristics, particularly the resistance against bacteria [41]. A significant increase in the antimicrobial activities of the treated jute fabric was recorded.

When the flame was applied to the untreated jute fiber, it burned within 1 min. However, the burned area decreased with the increasing chemical concentrations of T2 treatment in the following order T2F1 (0.045 ± 0.021) cm^2^ > T2F2 (0.03 ± 0.014) cm^2^ > T2F3 (0.015 ± 0.0007) cm^2^, indicating that the T2 treatment inhibited the growth of the flame in the jute fiber. Khatton et al. [33] also found that similar to this study, PO_4_ salt-treated jute fabrics inhibited the flame progression during the vertical flame test, where char length was measured. With an increase in chemical concentration, the char length decreased indicating that the flame-retardant characteristics were integrated within the jute fabric. It was found that 30% chemical concentration provided the best fire-retardant results. In this study, 30% chemicals-treated jute fiber showed the lowest burned area compared to the other concentrations. In another study, in order to incorporate flame-retardant characteristics in vinyl ester biocomposites, jute fiber was treated with a novel technique of microwave irradiations using an aqueous solution of magnesium nitrate and sodium hydroxide. The burning rate measured in a test revealed that by Mg^2+^-doping jute fibers, the flame retardancy of the biocomposite was marginally improved with a reduction in burning rate of 14.5% [42]. In another study, jute was treated with sodium metasilicate nonahydrate (SMSN) and its flame retardant and antimicrobial resistance characteristics were studied [43]. A significant decrease in burning rate and excellent antimicrobial property against both Gram positive and negative bacteria were observed with 2% SMSN in comparison with the control sample.

In the case of T3 treatment, the contact angle was higher for the untreated jute fiber (94.76° ± 1.04), whereas the contact angle decreased with the increasing chemical concentrations (T3W1 (47° ± 9.67) > T3W2 (42° ± 1.22) > T3W3 (24° ± 1.02). This evidenced inducing water-repellant characteristics in the T3-treated jute fibers. However, Demirci et al. found that the hydrophilicity of PVC coating can be controlled by allylamine plasma treatment as the contact angle increased with the treatment time [44]. More recently, it was demonstrated that a silica nano sol and a commercial water-repellent chemical were employed to impart hydrophobic characteristics on a bleached jute fabric [45].

### 3.2. FTIR Analysis of the Jute Fibers

The results demonstrated the effects of each chemical treatment on the bulk fiber characteristics. The standardized tests confirmed the modification of the bulk fiber characteristics by each of the chemical treatments. FTIR analysis provides us further understanding on what chemical changes are actually happening on the fiber during the treatments.

#### 3.2.1. Microstructural Characteristics of Untreated Jute Fiber

Jute fiber contains three main constituents: cellulose, hemicellulose, and lignin [19]. The structures of the constituents are shown in Figure 7.

In the case of cellulose and hemicellulose (repetition of cellulose with –C–O–C– group), there are C–C, C–O–C, and C=C bonds, which are weaker compared to the lignin (phenol group) as they are not a long group. For this reason, at a higher wavelength (low frequency), first cellulose and then hemicellulose is responsive to FTIR [19], whereas at lower wavelength, the lignin is responsive. This information can be explained with the help of FTIR transmittance peak. The results of the transmittance spectra as depicted by the transmittance (%) versus the wave number are presented in Figure 8 within the range of 4000–700 cm^−1^.

From Figure 8, it was clear that FTIR peaks of the raw and treated jute fibers lie in the single bond, triple bond, double bond, and finger print regions with the wavelength of 2500–4000, 2000–2500, 1500–2000, and 600–1500 cm^−1^, respectively [27]. Each FTIR spectrum showed a broad peak around 3600–3200 cm^−1^ followed by a peak around 1600–1300 cm^−1^, 1200–1000 cm^−1^, and 800–600 cm^−1^, indicating the existence of hydrate and hydroxyl (OH^−^) which were present in the cellulose and hemicellulose of the jute fiber [46]. These peaks (3600–3200 cm^−1^) lay in the single bond region [27]. Moreover, Bodîrlău and Teacă [47] suggested that a broad peak appeared in the range of 3600–3200 cm^−1^ wavelength due to the strong bond formation between foreign particles and cellulose, hemicelluloses, and lignin of the jute. From the figure, it was also apparent that within the wavelength range of 2000–2500 cm^−1^, there was no peak in the raw jute fiber, suggesting the absence of the triple bond. The locations of the bands for lignin in the fingerprint region were 1593 cm^−1^ and 1506 cm^−1^ for the aromatic skeletal vibrations, 1458 cm^−1^ and 1420 cm^−1^ for the C–H deformation, 1328 cm^−1^ for the syringyl ring plus guaiacyl ring, 1234 cm^−1^ for the syringyl ring and C=O stretch, and 1120 cm^−1^ for the aromatic skeletal vibrations [28]. Therefore, it could be stated that the lignin fingerprint region lay in lower wavelengths whereas cellulose and hemicellulose related to OH and another peak existed in the higher wavelengths. The summary of the observed transmittance peaks with the corresponding wavelengths and the shifting of the same peak is shown in Table 3.

#### 3.2.2. FTIR Spectra Analysis of Rot-Retardant Jute Fiber

FTIR spectra (Figure 9) were obtained using attenuated total reflectance (ATR) scanning for both raw and rot-retardant (T1)-treated jute fibers in order to confirm the chemical reaction between CuSO_4_ and cellulose backbone of the jute fiber. After treatment, there is a possibility of the formation of various compounds related to CuSO_4_. Therefore, peaks of S as well as SO_4_ were expected. According to the literature, the spectral bands appeared in the region of 1490–1410 cm^−1^, particularly at 1427 cm^−1^, and 880–860cm^−1^, particularly at 830 cm^−1^, which could be due to the formation of CO_3._ Similarly, the peaks formed in the regions of 1380–1350 cm^−1^, particularly at 1380 cm^−1^, and 840–815 cm^−1^, particularly at 830 cm^−1^, could represent S [27]. It should be noted that in the case of two consecutive peaks, the first transmittance peak was intense and the second one was weaker to medium strength and narrow. In addition, lignin was characterized by a peak at around 1740 cm^−1^ in the spectral band region of 1750–1700 cm^−1^. A peak 1090 cm^−1^ followed by 610 cm^−1^ could also evidence the existence of SO_4_ in the treated jute fibers.

The FTIR spectra characterized the presence of cellulose by –OH peak at higher wavelength (3600–3200 cm^−1^). As transmittance shifts were concentration-dependent, the O–H bond shifted rightwards (3416 cm^−1^ for T1R1, 3420 cm^−1^ for T1R2, and 3450 cm^−1^ for T1R3) as compared to the raw jute (3390 cm^−1^) and became wider due to the increase in rot-retardant (T1) chemical concentration. Cu, being a metal, cannot be identified in FTIR. Therefore, its functional groups were found with sulphate peak at 1415–1380 cm^−1^, sulfuric acid peak at 1350–1342 cm^−1^, sulfone peak at 1300–1350 cm^−1^, and sulphoxide peak at 1070–1030 cm^−1^. There was a clear peak for sulfoxide at around 1050 cm^−1^ and small peaks found between 1300–1350 cm^−1^ could be sulfone or sulfuric acid, which might have originated from CuSO_4_. Small peaks between 1380–1400 cm^−1^ suggested the presence of the sulfate group. This fact confirmed that the SO_4_ group of CuSO_4_ reacted with the cellulose and an increase in chemical concentrations increased the intensity of the O–H peak. Higher transmittance was observed in the region around 3400 cm^−1^ for the unmodified jute (by 12%) over the T1R3 jute fiber since more hydroxyl groups were present. Similarly, OH stretching, CH stretching, C–H bending, and C–C stretching were shifted with the increasing chemical concentrations of the T1 treatment [4]. It is worth mentioning here that it was not possible to distinguish all other peaks in the T1-treated jute fibers because of the weak transmission. Moreover, the shifting of any peak towards the right meant the distance between the molecules decreased, which caused an increase in density. This also indicated an increase in the fiber crystallinity. Therefore, it could be expected that rot-retardant treatment would increase the strength [49]. This finding supports the authors’ previous results [4] of tensile loading effect on the treated fiber with different chemical concentrations. The peaks obtained in the rot retardant-treated fibers are summarized in Table 4.

Meanwhile, the presence of Cu as a monolayer covering the adsorbent surface was evidenced resulting from the Cu salt treatment of the fiber [51]. Furthermore, due to the presence of Cu as an antimicrobial agent, the treated fibers would act as rot retardant materials. As the chemical treatment was conducted at room temperature, CuSO_4_ would not react with lignin but would react with the cellulose structure. The probable chemical bonding mechanism between the jute cellulose and T1 chemicals is shown in Figure 10.

The information of change in fiber crystallinity due to chemical treatment can be obtained from the relative comparisons of the FTIR peaks. For example, the OH peak of all treated jute fibers (3450 cm^−1^) was shifted rightwards (3416 cm^−1^ for T1R1, 3436 cm^−1^, for T1R2, and 3459 cm^−1^ for T1R3). This meant that the treated jute fibers became denser and had increased crystallinity as compared to the raw jute. The crystallinity trend can be summarized in the following ascending order: raw < T1R1 < T1R2 < T1R3. The intensity of the peaks gradually increased with T1 chemical concentration.

#### 3.2.3. FTIR Spectra Analysis of Fire-Retardant Jute Fiber

As NH_4_ and PO_4_ salts were used for the T2 treatment on jute, possible peaks were found for PO_4_, NH_4_, or N–H. Figure 11 shows FTIR spectra of fire-retardant-treated jute fibers at various chemical concentrations. A decreasing trend in the intensity of the O–H transmittance band at 3450–3400 cm^−1^ indicated that the hydroxyl group content in the treated fibers were reduced after the T2 treatment. However, it should be noted that the transmittance bands appeared at higher wavelengths in the treated fibers. This meant that a stronger bond was produced with the chemical treatment [4]. It was also noticeable that the peaks of the treated fibers became sharper compared to the untreated jute.

Transmittance in the single bond region (4000–2500 cm^−1^) was obtained from O–H stretching vibrations in hydroxyl, phenol, and carboxyl groups. The intensity of the O–H peak also increased due to the presence of more hydroxyl groups in fire retardant-treated jute fibers. In addition, the intensity of C–H stretching decreased (70% for T2F1, 56% for T2F2, and 33% for T2F3) with the fire-retardant chemical concentration. At approximately 1430 cm^−1^, N–H bending was observed, and this demonstrated that NH_4_ was present in the treated fibers. In addition, two new peaks were observed at 400 and 500 cm^−1^ in the treated fibers [52]. The transmittance at the lower band (1300–800 cm^−1^) represented the existence of ester [47], vinyl, and aromatic compounds. The mid-IR spectral range comprised of (i) intense transmittance in the region between 3620–3630 cm^−1^ caused by vibrations of hydroxyl groups and (ii) two N–H bond vibrations at 1430 cm^−1^ (N–H bending) and from 2800 to 3400 cm^−1^ (N–H stretching). Combination bands involving the O–H stretching mode and some other lower frequency modes occurred in a range from 1730 cm^−1^ to 2150 cm^−1^ [52]. The observed transmittance peaks with the wavelengths and the shifting of the same peak are shown in Table 5.

Ammonium ion as a peak appeared at 3250 cm^−1^ followed by 1430 cm^−1^ after the T2 treatment. The peak at 2350 cm^−1^ represented the N–H bond whereas the phosphate ion was indicated by a peak in the range of 1100–1000 cm^−1^. The other two wavelengths at 400 cm^−1^ and 500 cm^−1^ characterized aliphatic iodo compounds (C–I) as a result of the T2 treatment.

Though similarities in the peak wavelength of the T2-treated jute fibers were noticed, their intensity varied with the chemical concentration. The O–H peak (3450 cm^−1^ for the raw jute fiber) shifted right for all T2-treated jute fibers (3250 cm^−1^). Similarly, C–H peak (2900 cm^−1^ for the raw jute fiber) remained in the same wavelength after treatment, but the intensities increased with chemical concentrations (0.86% for T2F1, 67% for T2F2, and 109% for T2F3 compared to raw jute fiber). With the increase in chemical concentration, N–H peak intensity increased, indicating the presence of N–H being prominent. Furthermore, the peak shifting was absent suggesting that the crystallinity of the fiber was not affected due to the fire-retardant treatment.

#### 3.2.4. FTIR Spectra Analysis of Water-Retardant Jute Fiber

The FTIR transmission peaks of the T3 treated jute fiber are shown in Figure 12. Due to the T3 treatment, it could be expected that FTIR spectra would contain various peaks such as Cl, vinyl, and polymer, representing the formation of various compounds of poly vinyl chloride. From the figure it is clear that in the water retardant-treated jute fibers, the O–H peak became broader with the increase in chemical concentration as the transmittance maximum value for O–H stretching was the concentration-dependent nature of the solvent and temperature [53].

The intensity of OH peaks in the water retardant-treated jute fibers decreased compared to the raw jute fiber. A higher transmittance peak was observed in the region around 3400 cm^−1^ for the raw jute fiber compared to the water-retardant jute fibers since more hydroxyl groups were present. The peak intensity at 2900 cm^−1^ was not only increased with the T3 chemical concentrations but the shape of the peak also changed. The ascending order of C–H intensity (T3W1 < T3W2 < T3W3) in the spectra meant that the bond became prominent with the PVC treatment. Furthermore, in the triple bond region (2500–2000 cm^−1^), a small peak appeared and its intensity gradually increased with the increase of T3 chemical concentration as compared to the raw jute fiber. A similar scenario was also observed in the double bond region (2000–1500 cm^−1^), fingerprint region, and for peaks at some specific wavelengths (1420 cm^−1^, 1250 cm^−1^, and 800 cm^−1^). The observed transmittance peaks with the wavelengths are listed in Table 6.

The FTIR spectra also revealed a peak of vinyl C–H in plane at 1420–1410 cm^−1^, more specifically, 1420 cm^−1^ in the finger print region. Furthermore, Cl as compound and vinyl C–H in plane bend and out of plane bend in the jute fiber after T3 treatment were indicated by a peak at 1850 cm^−1^ and a broader peak at 800 cm^−1^. Moreover, the presence of vinyl-related compound was evidenced by the peak at 1000 cm^−1^.

### 3.3. FTIR Analysis of the Jute Fiber-Reinforced Composite

The characteristics of a fiber-reinforced composite are mostly dependent on fiber type, strength, modulus, length and orientation, fiber/matrix interfacial bonding, and fiber content [54]. More specifically, internal structure as well as the chemical constituents of the fiber will determine the fiber strength characteristics. Eventually, the properties of the fiber and matrix materials and their interaction would govern the performance of the composite end product. Thus, it is absolutely important to identify the chemical constituents and their structures in the components of the composite end product. FTIR can help to conduct this type of analysis for the fiber, matrix, and compound materials such as composite.

The effects of fiber surface treatments were discussed by Shahinur et al. [4]. Furthermore, Shahinur et al. [55] revealed that the chemical treatments affected the mechanical properties. It was observed that the tensile properties of the raw and treated jute fibers changed with the chemical concentration [56]. Similar results were also reported by Ben Brahim and Ben Cheikh [57], when the effect of volume fraction of fibers on the tensile properties (longitudinal modulus and the longitudinal stress) of unidirectional alfa-polyester composites was studied. In this study, representative treated fibers and their reinforced composites were selected and analyzed with the FTIR to identify the change in jute structure due to the chemical treatments on the fiber-reinforced composite.

Figure 13 presents the FTIR spectra of the composites. The O-H bond of the jute fibers as well as their associated composites occurred as ~3450 cm^−1^. The intensity of the OH bond increased in the composites made with the treated jute fibers compared to the composite made with the raw jute. The results also revealed that the relation between the OH bond and the percentage of transmittance could be represented in following ascending order JT2-PP < JT3-PP < JT1-PP at 3450 cm^−1^. A similar trend was also found for the second peak at 2920 cm^−1^ representing C–H bond which gradually increased for JT1-PP and JT3-PP, whereas it remained same for the JT2-PP.

Peaks at wavenumbers 2958, 2920, 2839, 1458, 1377, and 719 cm^−1^ represented the characteristics of PP [40]. Furthermore, a peak at ~1744 cm^−1^ indicated the symmetric C≡O stretching of anhydride functions grafted on PP [56].

Meanwhile, the peaks in the raw jute composites indicated the presence of cellulose, hemicellulose, and lignin. Furthermore, for the treated jute composites, some extra peaks represent the effect of the treatments. Rot-retardant jute composite (JT1-PP) indicated the presence of SO_4_, S, and CO_3_, which could be related to the peaks at 1090/610 (sharp peak/broad peak) cm^−1^, 1830/830 cm^−1^, and 1427/830 cm^−1^, respectively. Meanwhile, the N–H bond was available in the JT2-PP at 2350 cm^−1^. Furthermore, peaks at 3250/1430 cm^−1^ and 1100 cm^−1^ in JT2-PP also indicated the presence of NH_3_ and PO_4_, respectively. The presence of vinyl in the JT3 was confirmed by the 1420/1850 cm^−1^ peak becoming sharper compared to the J-PP. Last but not least, the peak availability at 800 cm^−1^ represented the presence of Cl in the JT3-PP.

From Figure 11, it can be demonstrated that the FTIR spectra found for the composites are the superimposed version of the respective treated fibers as well as the matrix. Therefore, it can be safely assumed that the composite products should display the retardant behaviors similar to the treated jute fibers. When the FTIR spectra of jute fiber and composite were compared, no indication of degradation was observed in the composites.

### 3.4. Results Summary

The major findings in terms of incorporating the retardant characteristics in the jute fiber for developing jute polymer composites are summarized in Table 7. The presence of different chemicals identified by FTIR evidenced the success of treatments. Further physical evidences on changing the bulk characteristics of the fibers demonstrated the effectiveness of the treatments. The retardant mechanisms of the treated fiber need further investigation. In future, further work has been planned to identify the effects of different chemical treatments on the biodegradability and retardant criteria of the jute-based composites. Though for evaluating the rot retardant characteristics, only bacterial activity was tested, their fungus activity needs to be explored in future for better clarification. The primary focus in this work was to identify the effect of individual treatment on the jute fiber. However, a combination of different treatments on the retardant characteristics could be evaluated as an extension of this study in future.

## 4. Conclusions

A comprehensive FTIR study was carried out on untreated and treated jute fibers and their reinforced composites. Different chemicals were employed to conduct rot- (T1), fire- (T2), and water- (T3) retardant treatments. This study notably contributed to the knowledge and understanding of the chemical effects on the jute for a jute-based product development which will influence industrial and academic jute research. The following conclusions can be drawn based on the observations made in this study.

Raw and treated jute fibers contain cellulosic and hemicellulose OH. FTIR studies confirmed the impregnation of SO_4_, in the case of T1 treatment, PO_4_ and NH_3_ in the case of T2 treatment, and Cl and vinyl in the case of T3 treatment.

Rot-, fire-, and water-retardant characteristics were demonstrated in the treated jute fibers by higher bacteria inhibition zone, lower burning rate, and lower contact angle in comparison with the untreated jute fibers.

In each case of the chemical treatments, increasing concentration of the chemicals during the treatments resulted in improved retardant characteristics in the jute fibers.

For rot retardant, 8% concentration is better, for fire retardant 30% concentration is better, and for water retardant, 20% is better based on performance and cost of the chemicals.

In the composites developed using the treated fibers, it was found that the main ingredients of the chemicals used for the jute fiber treatments, such as Cu, SO_4_, NH_3_, PO_4_, vinyl, and Cl, were also present, indicating that the composites could retain the retardant characteristics similar to the treated jute.

Polymer composites made with different retardant-treated jute fibers could broaden their industrial applications.

## Figures and Tables

**Figure 1 polymers-13-02571-f001:**
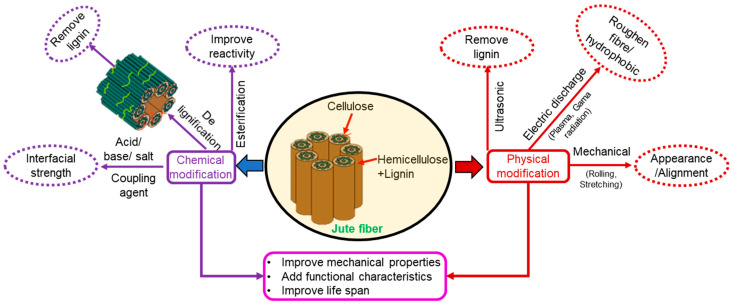
Overview of treatments on the jute fiber for polymer composite fabrication.

**Figure 2 polymers-13-02571-f002:**
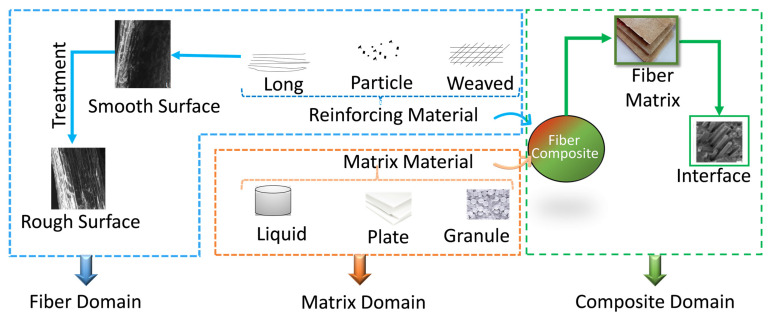
Different forms of matrix and jute fiber reinforcing materials.

**Figure 3 polymers-13-02571-f003:**
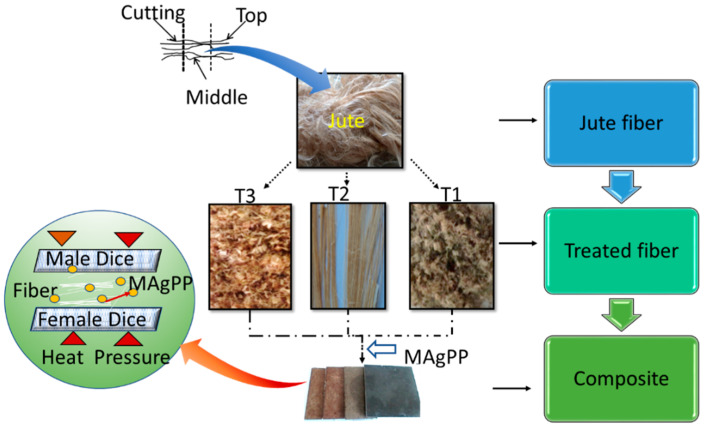
Schematic diagram of workflow for fiber treatment, composite fabrication, and characterization.

**Figure 4 polymers-13-02571-f004:**
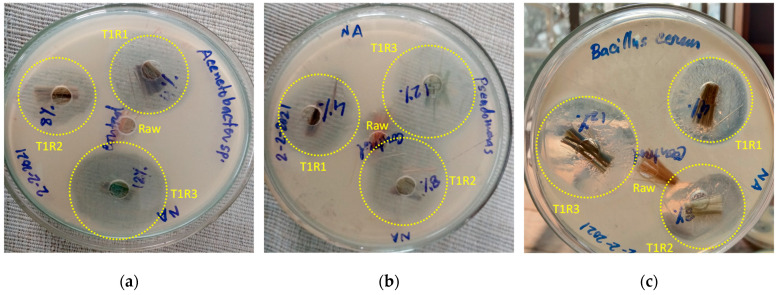
Inhibition zone diameters caused by the rot retardant jute fibers against (**a**) *Acinetobacter* sp., (**b**) *Pseudomonas* sp., (**c**) *Bacillus cereus*, (**d**) *Salmonella* sp., and (**e**) *E. coli*.

**Figure 5 polymers-13-02571-f005:**
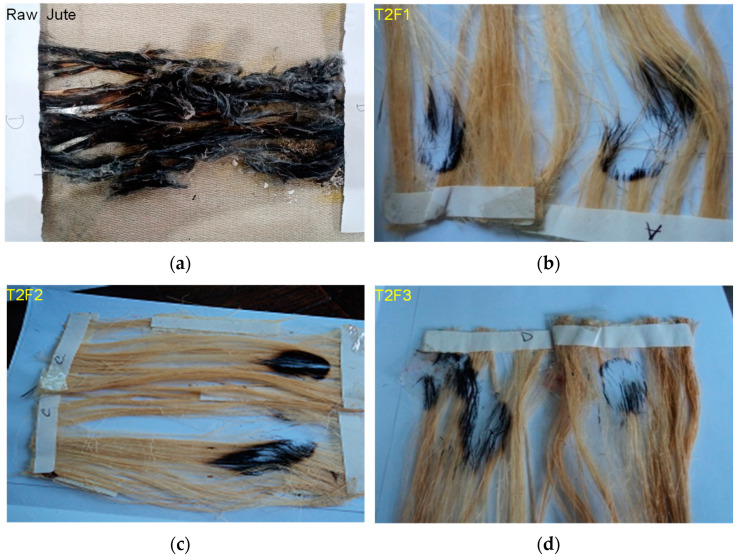
Evidence of fire tests on (**a**) raw jute and treated fibers with different concentrations, (**b**) 20 wt.%, (**c**) 25 wt.%, and (**d**) 30 wt.%.

**Figure 6 polymers-13-02571-f006:**
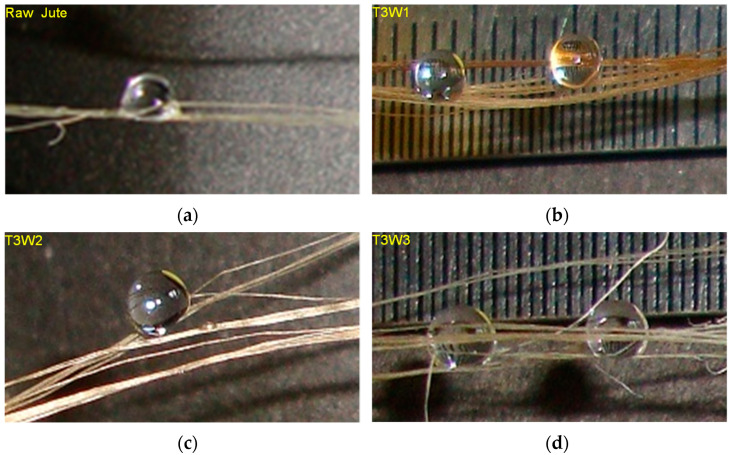
Contact angle measurements on (**a**) raw jute and treated fibers with different concentrations, (**b**) 10 wt.%, (**c**) 15 wt.%, and (**d**) 20 wt.%.

**Figure 7 polymers-13-02571-f007:**
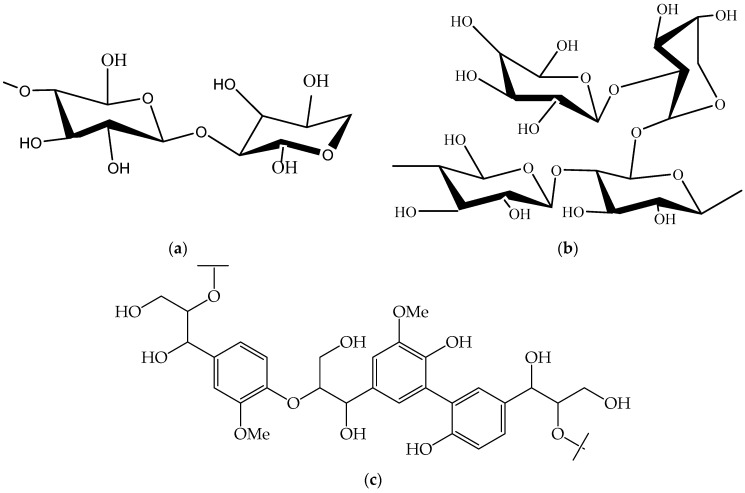
Structure of jute (**a**) cellulose, (**b**) hemicellulose, and (**c**) lignin.

**Figure 8 polymers-13-02571-f008:**
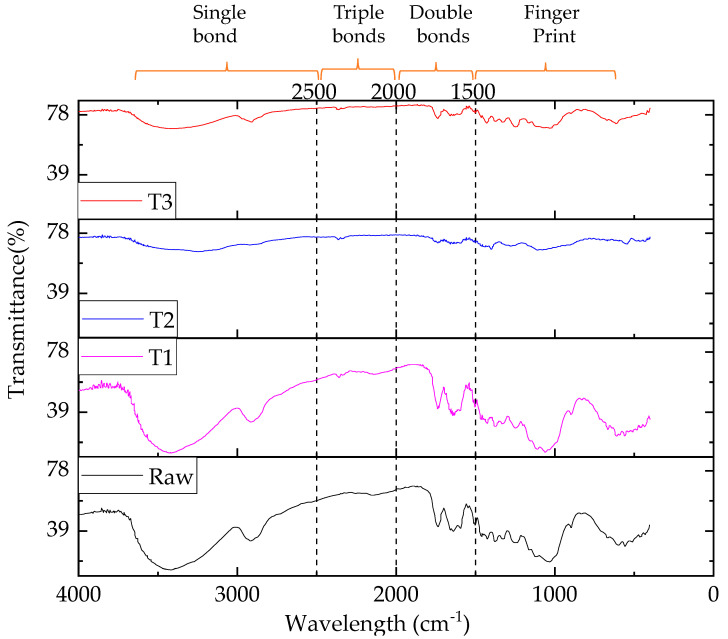
FTIR-transmittance spectra and bond regions of raw and treated jute fibers (T1: Rot retardant, T2: fire retardant, and T3: water retardant).

**Figure 9 polymers-13-02571-f009:**
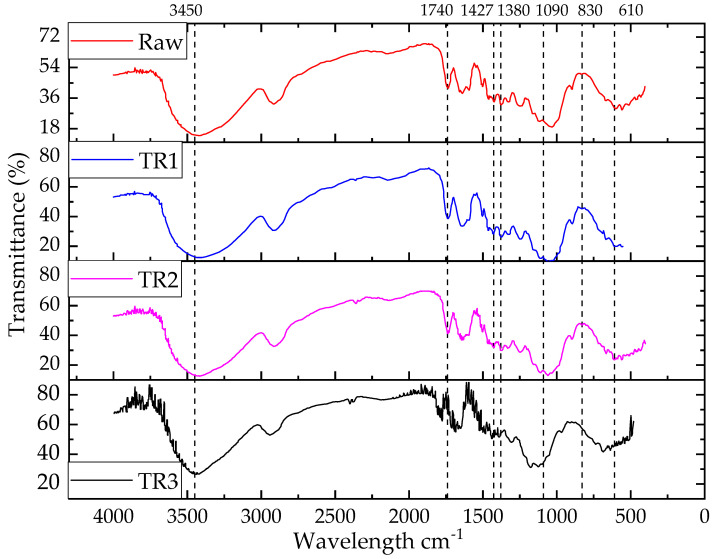
FTIR transmission spectra of rot-retardant (T1)-treated jute fibers.

**Figure 10 polymers-13-02571-f010:**
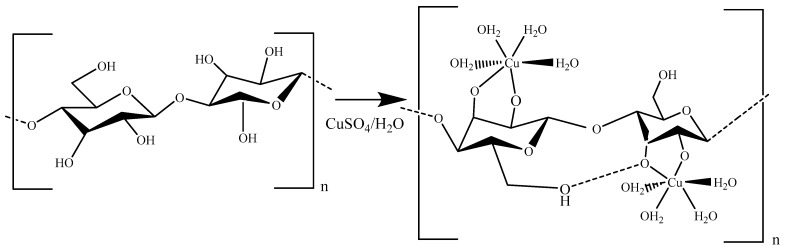
Possible mechanism of bond formation between CuSO_4_ and jute cellulose.

**Figure 11 polymers-13-02571-f011:**
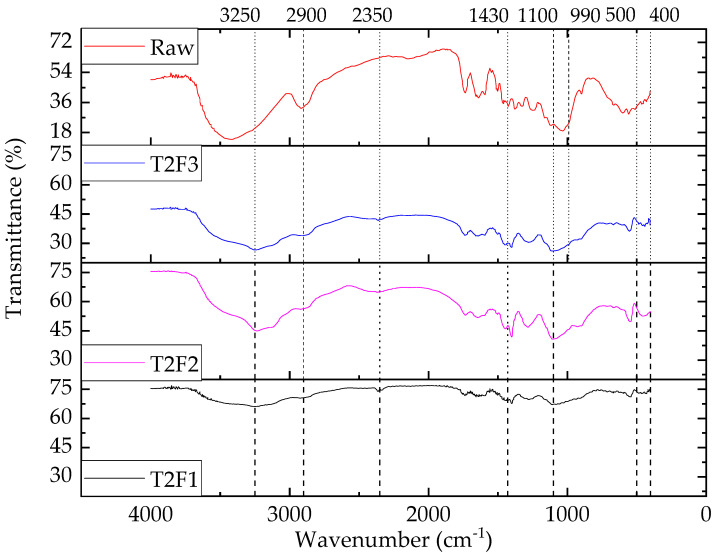
FTIR transmission spectra of fire retardant (T2)-treated jute fibers.

**Figure 12 polymers-13-02571-f012:**
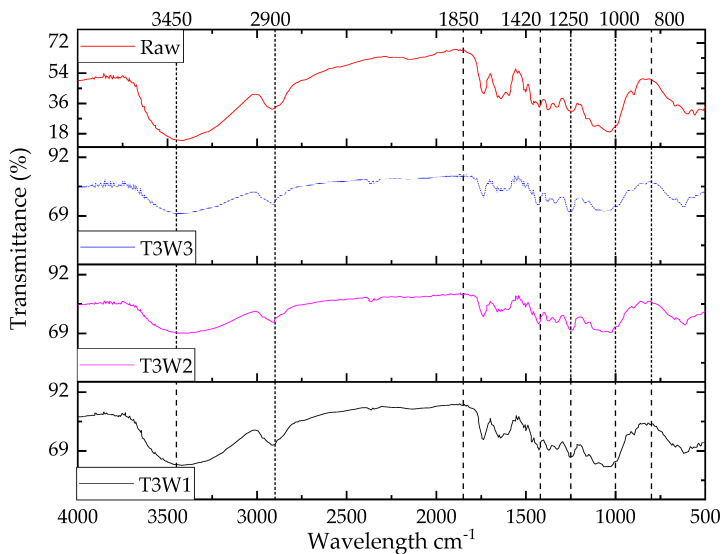
FTIR transmission spectra of water retardant (T3)-treated jute fibers.

**Figure 13 polymers-13-02571-f013:**
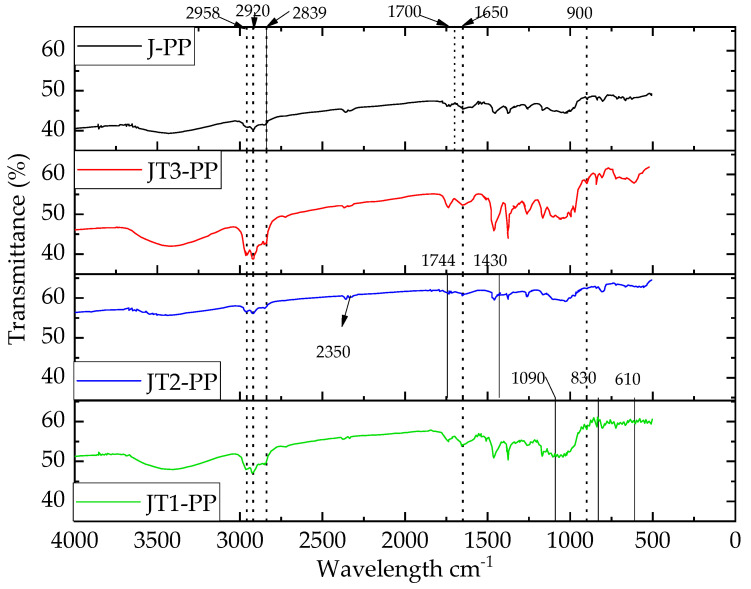
FTIR of MAgPP composites reinforced with chemically modified jute fiber (25 wt.%).

**Table 1 polymers-13-02571-t001:** Summary of jute and fiber-reinforced composite samples with different chemical treatments.

Sample Code		Name of Treatment	Chemical Name	Physical Form	Molecular Weight (gm/mol)	Chemical Concentration (wt.%)	Sample Colour	Composite
T0		-	-			-	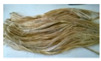 Light golden	J-PP
T1	T1R1	Rot retardant	CuSO_4_·5H_2_O	Blue granule	249.68	4	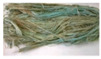 Light bluish	JT1-PP
	T1R2					8		
	T1R3					12		
T2	T2F1	Fire retardant	(NH_4_)_2_·HPO_4_	White granule	132.06	20	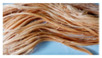 Bright golden	JT2-PP
	T2F2					25		
	T2F3					30		
T3	T3W1	Water retardant	([C_2_H_3_Cl]_n_ or PVC	White powder	48,000	10	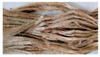 Bright brown	JT3-PP
	T3W2					15		
	T3W3					20		

**Table 2 polymers-13-02571-t002:** Diameters (mm) of inhibition zones against different bacteria.

Bacteria/Strain	T1R1	T1R2	T1R3
*Acinetobacter* sp.	14.5 ± 0.6	14.4 ± 0.2	19.9 ± 0.4
*Pseudomonas* sp.	15.2 ± 0.48	19.3 ± 0.04	24.6 ± 0.28
*Salmonella* sp.	0	0	0
*E. coli*	0	0	12.2 ± 0.42
*Bacillus cereus*	17.8 ± 0.06	20.7 ± 0.02	20.9 ± 0.05

**Table 3 polymers-13-02571-t003:** Summary of important FTIR transmittance peaks in untreated jute fibers.

Band Position (cm^–1^)	Functional Group	Ref.
~3600–3200	ν(OH) broad, strong bond from the cellulose, hemicelluloses, and lignin of jute	[46,48]
~3000–2900	ν(C–H) in aromatic ring and alkenes	
~2930–2910	C–H methyl and methylene groups	
~1740–1730	C= O carbonyls	
~1750–1710	ν(C=O) most probably from the lignin and hemicelluloses	
~1650–1630	Possibly aromatic ring	
~1640–1618	C=C alkenes	
~1630–1642.6	Probably absorb water	
~1515–1504	ν(C=C) aromatic in plane	
~1501–1510	ν(C=C) aromatic skeletal ring vibration due to lignin	
~1462–1425	CH_2_ cellulose, lignin	
~1460–1468	δ (C–H: C–OH) 1^o^ and 2^o^ alcohol	
~1422–1428	δ (C–H)	
~1384–1346	C–H cellulose, hemicelluloses	
~1365–1377	δ(C–H)	
~1315	δ(C–H)	
~1280	δ(C–H_2_) twisting	
~1260–1234	O–H phenolic	
~1170–1153	O–H alcohols (primary and secondary) and aliphatic ethers	
~1155	ν(C–C) ring breathing, asymmetric	
~1112	ν(C–O–C) glycosidic	
~1055	ν(C–O–C) 2^o^ alcohol	
~1033	ν(C–O–C) 1^o^ alcohol	
~910	C=C alkenes	
~895	ν(C–O–C) in plane, symmetric	

**Table 4 polymers-13-02571-t004:** Summary of important FTIR transmittance peaks in rot retardant (T1)-treated jute fibers.

Wavelength (cm^−1^)	Peak Details		Reference
3600–3200	O-H		[27]
1700–1750	Lignin and hemicellulose		[46]
1500–1100	Lignin fingerprint		[50]
1130–1080/680–610 ^5^	Sulphate ion (SO_4_)		[27]
1380–1350/840–815 ^5^	Sulfur		
1050 1300–1350 1380–1400	Sulfate group Sulfoxide Sulfone/Sulfuric acid	SO_4_	[Spectra Base™, Wiley. https://spectrabase.com/ accessed on 25 July 2021]

Note: 5 stands for the first transmittance is intense and broad and the second has weak to medium intensity and narrow.

**Table 5 polymers-13-02571-t005:** Summary of important FTIR transmittance peaks in fire retardant (T2)-treated jute fibers.

Wavelength (cm^−1^)	Peak Details	Reference
3620–3630	OH	[52]
3300–3030/1430–1390^5^	Ammonium ion	[27]
2800 to 3400	N–H stretching	
2350	N–H	
1730 to 2150	O–H stretching	
1430	N–H bending	
1100–1000	Phosphate ion	
1050–990	Aliphatic phosphate (P–O–C stretch)	
400	Aliphatic iodo compounds C-I	
500		

Note: 5 stands for the first transmittance which is intense and broad and the second has weak to medium intensity and narrow.

**Table 6 polymers-13-02571-t006:** Summary of important FTIR peaks in water retardant (T3)-treated jute fibers.

Wavelength (cm^−1^)	Peak Details		Reference
1420–1410	Vinyl C–H in plane bend Vinyl C–H out of plane bend	Functional group	[27]
995–985/915–890			
1300–800	Esterification		
1000	Vinyl-related compound		
800–700	Aliphatic chloro compound, C–Cl stretch		

**Table 7 polymers-13-02571-t007:** Summary of findings from different treatments (T1, T2, and T3) on the jute fibers.

Treated Jute Fiber Features	Rot-Retardant Fiber (T1)	Fire-Retardant Fiber (T2)	Water-Retardant Fiber (T3)
Cellulosic and hemicellulose OH	Present and shifted right	Present and shifted right	Present
Characteristic peak	SO_4_	NH, PO_4_	Cl, vinyl
Crystallinity	Increased with chemical concentration	Increased with chemical concentration	Increased with chemical concentration
Change in level of retardant characteristics	Present and increased with chemical concentration	Present and increased with chemical concentration	Present and increased with chemical concentration
Possible evidences of retardant characteristics	Increasing inhibition zone with chemical concentration	Decreasing burning area	Decreasing contact angle

## Data Availability

The data presented in this study are available within the article.

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
