# Peer review of "Effect of Rot-, Fire-, and Water-Retardant Treatments on Jute Fiber and Their Associated Thermoplastic Composites: A Study by FTIR"

_polymers, 2021, doi:10.3390/polym13152571_

Round 1

Reviewer 1 Report

  1. In lines 42-43, the authors say “it is an annual crop … which has 0% recycle fraction”. Can the authors explain what they will say about recycle fraction?
  2. Lines 49, 55, 118, 132, and 446. Some references are missed “Error! Reference source not found”.
  3. The color of de composites after compression molding seems dark. Is some degradation due to the temperature and time of processing? Can the authors verify by its analysis?
  4. The authors can give the MAgPP content? Is the coupling agent optimized? How effects the coupling agent to the rot, fire, and water composites behavior?
  5. The chemical concentrations of the three treatments are different. The authors can give an explanation or bibliography to justify these chemical amounts?
  6. The authors have shown images of the fiber test (example: figure 5). Are these tests also performed on the composite?
  7. The authors have applied the combination of the different treatments?

Reviewer 2 Report

In Table 1 instead of describing the color please just present it.

Why such a concentrations of modifiers were applied? Any previous results? Please mention them if so.

Please characterize the applied modifiers, especially PVC applied, what type, molecular weight, etc.

Why Authors did not apply extruder or at least some internal mixer to prepare composites? Described procedure does not guarantee proper distribution of filler.

Please compare the results on the rot modification with the literature data.

Same for flame retardancy.

And for water resistant treatment.

Please present all FTIR specta with the same scale on Y axis so they could be compared.
